# A Graph Similarity for Deep Learning

**Seongmin Ok**
Samsung Advanced Institute of Technology
Suwon, South Korea
seongmin.ok@gmail.com

## Abstract

Graph neural networks (GNNs) have been successful in learning representations from graphs. Many popular GNNs follow the pattern of *aggregate-transform*: they aggregate the neighbors' attributes and then transform the results of aggregation with a learnable function. Analyses of these GNNs explain which pairs of non-identical graphs have different representations. However, we still lack an understanding of how similar these representations will be. We adopt kernel distance and propose *transform-sum-cat* as an alternative to aggregate-transform to reflect the continuous similarity between the node neighborhoods in the neighborhood aggregation. The idea leads to a simple and efficient graph similarity, which we name Weisfeiler–Leman similarity (WLS). In contrast to existing graph kernels, WLS is easy to implement with common deep learning frameworks. In graph classification experiments, transform-sum-cat significantly outperforms other neighborhood aggregation methods from popular GNN models. We also develop a simple and fast GNN model based on transform-sum-cat, which obtains, in comparison with widely used GNN models, (1) a higher accuracy in node classification, (2) a lower absolute error in graph regression, and (3) greater stability in adversarial training of graph generation.

## 1 Introduction

Graphs are the most popular mathematical abstractions for relational data structures. One of the core problems of graph theory is to identify which graphs are identical (i.e. *isomorphic*). Since its introduction, the Weisfeiler–Leman (WL) algorithm (Weisfeiler & Leman, 1968) has been extensively studied as a test of isomorphism between graphs. Although it is easy to find a pair of non-isomorphic graphs that the WL-algorithm cannot distinguish, many graph similarity measures and graph neural networks (GNNs) have adopted the WL-algorithm at the core, due to its algorithmic simplicity.

The WL-algorithm boils down to the *neighborhood aggregation*. One of the most famous GNNs, GCN (Kipf & Welling, 2017), uses degree-normalized averaging as its aggregation. GraphSAGE (Hamilton *et al.* , 2017) applies simple averaging. GIN (Xu *et al.* , 2019) uses the sum instead of the average. Other GNN models such as GAT (Veličković *et al.* , 2018), GatedGCN (Bresson & Laurent, 2017), and MoNet (Monti *et al.* , 2017) assign different weights to the neighbors depending on their attributes before aggregation.

All the methods mentioned above follow the pattern of *aggregate-transform*. Xu *et al.* (2019) note that many GNNs based on graph convolution employ the same strategy: aggregate first and then transform. In this paper, we identify a problematic aspect of aggregate-transform when applied to graphs with continuous node attributes. Instead, we propose *transform-sum-cat*, where *cat* indicates concatenation with the information from the central node. We justify our proposal by applying the well-established theory of kernel distance to the WL algorithm. It naturally leads to a simple and fast graph similarity, which we name *Weisfeiler–Leman similarity* (WLS).

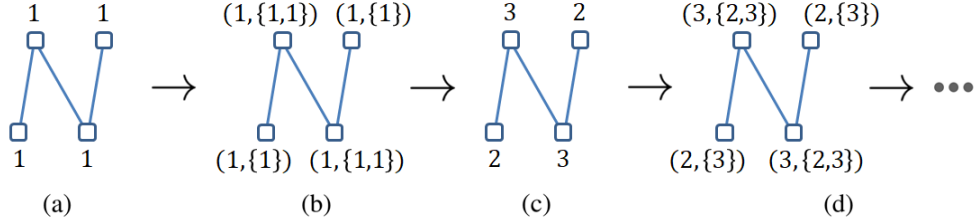

Figure 1: Illustration of WL-iterations. (a) We set $f(v) = 1$ for all $v \in V(G)$ initially, if not given in the data. (b) Each node attribute is updated with the pair of itself and the (multi)set of neighbor attributes. (c) The attributes are re-labeled for the convenience of further iterations. (d) Steps (b) and (c) are repeated for a fixed number of iterations. See Section 2.2.

We test the applicability of our proposal in several experiments. First, we compare different aggregation methods from GNN literature for graph classification where transform-sum-cat outperforms the rest. Then, we build a simple GNN based on the same idea, which obtains (1) a higher accuracy than other popular GNNs on node classification, (2) a lower absolute error on graph regression and when used as a discriminator, (3) enhanced stability of the adversarial training of graph generation. We summarize our contributions as follows.

- We propose a transform-sum-cat scheme for graph neural networks, as opposed to the predominantly adopted aggregate-transform operation. We present examples where transform-sum-cat is better than aggregate-transform for continuous node attributes.

- We define a simple and efficient graph similarity based on transform-sum-cat, which is easy to implement with deep learning frameworks. The similarity extends the Weisfeiler–Leman graph isomorphism test.

- We build a simple graph neural network based on transform-sum-cat, which outperforms widely used graph neural networks in node classification and graph regression. We also show a promising application of our proposal in one-shot generation of molecular graphs.

The code is available at `https://github.com/se-ok/WLsimilarity`.

## 2 Preliminaries

### 2.1 Notations

Let $G$ be a graph with a set of nodes $V(G)$ or simply $V$. We assume each node $v \in V$ is assigned an *attribute* $f(v)$, which is either a categorical variable from a finite set or a vector in $\mathbb{R}^d$. If we update the attribute on $v$, the original attribute is written as $f^0(v)$ and the successively updated ones as $f^1(v), f^2(v)$, etc. The set of nodes adjacent to $v$ is denoted as $\mathcal{N}(v)$. The edge that connects $u$ and $v$ is denoted as $uv$. We denote the concatenation operator as $\oplus$. Abusing the common notation, we shall write a *multiset* simply as a *set*.

### 2.2 Weisfeiler–Leman isomorphism test

The Weisfeiler–Leman (WL) test (Weisfeiler & Leman, 1968) is an algorithmic test of isomorphism (identity) between graphs, possibly with categorical node attributes. Although the original test is parameterized by dimension $k$, we only explain the 1-dimensional version, which is used in most machine learning applications.

Let $G$ be the path of length 3 depicted in Figure 1 (a). If $G$ has no node attributes we set $f(v) = 1$ for all $v \in V(G)$. Then, we update the node attributes in "stages," once for all nodes at each stage. The updated attribute is the pair of itself and the set of attributes of its neighbors. In Figure 1 (b), the middle vertices have two 1's in the (multi)set notation { }. For further iterations, the new attributes may be re-labeled via an injective mapping, for example, $(1, \{1\}) \to 2$ and $(1, \{1, 1\}) \to 3$, as in Figure 1 (c). The next iteration is done by analogy, as in Figure 1 (d).

After a fixed number of iterations, we compare the set of resulting attributes to that from another graph. If two sets differ, then the two graphs are non-isomorphic and are *distinguishable* by the

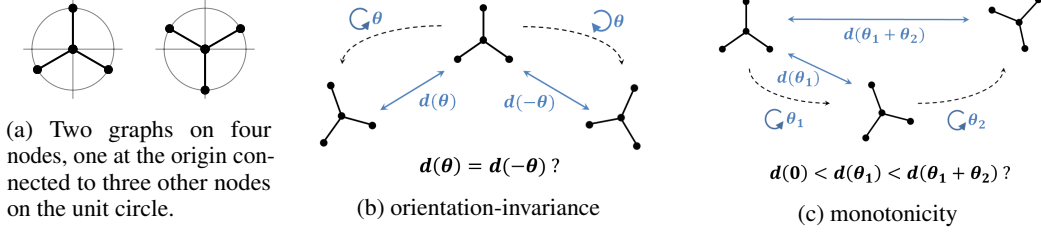

(a) Two graphs on four nodes, one at the origin connected to three other nodes on the unit circle.

$d(\theta) = d(-\theta)$ ?

(b) orientation-invariance

$d(0) < d(\theta_1) < d(\theta_1 + \theta_2)$ ?

(c) monotonicity

Figure 2: Illustration of (a): a problem of existing neighborhood aggregation methods, and (b), (c): desirable properties of neighborhood distance for continuous attributes. The two graphs in (a) have 2D coordinates as the node attributes. The neighborhood aggregation at the central nodes are indistinguishable by common aggregation methods. We claim that a good neighborhood representation should have an associated distance that is (b) orientation-invariant, and (c) strictly increasing up to the degree of small rigid transformation. See Section 2.4 for further discussion.

WL-test. Many indistinguishable non-isomorphic pairs of graphs exist. However, asymptotically, the WL-test uniquely identifies almost all graphs; c.f. Babai *et al.* (1980); Arvind *et al.* (2017).

## 2.3 Vector representation of a set

In this section, we briefly introduce the kernel distance between the point sets, focusing only on what is required in this paper. To summarize, we represent a set of vectors by the sum of the vectors after applying a transformation called a *feature map*. For an excellent introduction to the kernel method, see Phillips & Venkatasubramanian (2011) or Hein & Bousquet (2004).

Let $K : \mathbb{R}^d \times \mathbb{R}^d \to \mathbb{R}$ be a function. For example, we may think of the Gaussian kernel $\exp\big(-\frac{\|x-y\|^2}{2}\big)$. Function $K$ is a *positive definite kernel* (*pd kernel*) if, for any constants $\{c_i\}_{i=1}^n$ and points $\{x_i\}_{i=1}^n$ in $\mathbb{R}^d$, we have $\sum_i \sum_j c_i c_j K(x_i, x_j) \geq 0$. A pd kernel $K$ has an associated reproducing kernel Hilbert space $\mathcal{H}$ with feature map $\phi : \mathbb{R}^d \to \mathcal{H}$ such that $K(x, y) = \langle \phi(x), \phi(y) \rangle$ for all $x, y \in \mathbb{R}^d$, where $\langle , \rangle$ denotes the inner product on $\mathcal{H}$.

A pd kernel $K$ with feature map $\phi$ induces a (pseudo-)distance $d_K$ on $\mathbb{R}^d$, which is defined by $d_K^2(x, y) = \|\phi(x) - \phi(y)\|_{\mathcal{H}}^2 = \langle \phi(x) - \phi(y), \phi(x) - \phi(y) \rangle = K(x, x) - 2K(x, y) + K(y, y)$. For sets of points $X = \{x_i\}_{i=1}^m$ and $Y = \{y_j\}_{j=1}^n$, the induced distance $D_K$ is similarly defined as

$$D_K^2(X, Y) = \sum_{x \in X} \sum_{x' \in X} K(x, x') - 2 \sum_{x \in X} \sum_{y \in Y} K(x, y) + \sum_{y \in Y} \sum_{y' \in Y} K(y, y')$$
$$= \big\| \sum_{x \in X} \phi(x) - \sum_{y \in Y} \phi(y) \big\|^2.$$

Hence, $\phi(X) = \sum_i \phi(x_i)$ represents set $X$ independently of $Y$, and the set distance $D_K$ can be computed using the distance between the representation vectors. If points $x_i$ and $y_j$ are associated with weights $v_i$ and $w_j$ in $\mathbb{R}$, then we replace $K(x_i, y_j)$ with $v_i w_j K(x_i, y_j)$ and obtain $\phi(X) = \sum_i v_i \phi(x_i)$ and $\phi(Y) = \sum_j w_j \phi(y_j)$ in the same manner. For many known kernels, the explicit feature maps are unclear or infinite-dimensional. However, we remark that when walking backward, for an arbitrary map $\phi : \mathbb{R}^d \to \mathbb{R}^D$, there is an associated set similarity using the representation map $\phi(X) = \sum_{x \in X} \phi(x)$. Its usefulness depends on specific problems at hand. For instance, in Section 3.2, we define a neural network based on WLS, where $\phi$ *learns* a suitable feature map in a supervised task.

## 2.4 Problems of aggregation-first scheme

Many modern GNNs can be analyzed with the Weisfeiler–Leman framework (Xu *et al.*, 2019). It is common practice to aggregate the attributes of neighbors first and then transform with a learnable function. However, we do not yet have a reliable theory that explains this order of operations. Instead, we present two examples where aggregation-first might be dangerous for continuous node attributes.

Let us consider a graph on four nodes drawn on the Euclidean plane: one node at $(0, 0)$ connected to three other nodes on the unit circle. See Figure 2 (a) for two different examples. We assign the

2D coordinates as the node attributes. If we apply the common neighborhood aggregation at the central nodes (such as average, sum, or various weighted sums), the results of the two graphs are indistinguishable. However, if we apply the set representation $\sum \phi(x)$ from Section 2.3 with $\phi$ from a polynomial kernel, not only can we distinguish the two graphs but we can also deduce the exact locations of three neighbors from the set representation. See Appendix B.2 for a detailed explanation.

We target two properties, illustrated in Figure 2, for a good neighborhood representation. We consider the (dis-)similarity between a set of vectors and another set obtained by applying a rigid transformation to all elements of the former set. First, *orientation-invariance* indicates the following: (1) the similarity from a rotation should be the same as the similarity from the inverse of the rotation. (2) the similarity from a translation to a direction should be the same as the similarity from the same amount of translation to another direction. Second, *monotonicity* indicates that, when we apply a rotation or a translation, the dissimilarity must be strictly increasing up to a small positive degree of transformation. Note that the set representation from Section 2.3 with the Gaussian kernel satisfy both orientation-invariance and monotonicity. However, fast kernels and popular GNN aggregations have difficulties. For formal definition and discussion, see Appendix B.3.

## 3 Weisfeiler–Leman similarity

In this section we define Weisfeiler–Leman Similarity (WLS), which describes a diverse range of graph similarities rather than a single reference implementation. After introducing the general concept, we provide an implementation of WLS as a graph kernel in Section 3.1. We also present a similarly designed neural network in Section 3.2.

Recall from Section 2.2 that the core idea of the WL-algorithm is to iteratively update the node attributes using the neighbors' information. Our focus is to reflect the similarity between the sets of neighbors' attributes into the node-wise updated attributes via the set-representation vector from Section 2.3. A formal statement is in Algorithm 1.

---

**Algorithm 1:** Updating node attributes in Weisfeiler–Leman similarity

---

**Data:** Graph $G$, nodes $V$, initial attributes $f^0(v)$ for $v \in V$, iteration number $k$, and feature
   maps $\phi_i$ for $i = 1, 2, \ldots, k$.
**Result:** Updated attributes $f^k(v)$ for $v \in V$.
**for** $i \leftarrow 1$ **to** $k$ **do**
   **for** $v \in V$ **do**
      $g^i(v) \leftarrow \phi_i(f^{i-1}(v))$;
      $\hat{f}^i(v) \leftarrow \sum_{u \in \mathcal{N}(v)} g^i(u)$;
      $f^i(v) \leftarrow \text{COMBINE}_i \left( f^{i-1}(v), \hat{f}^i(v) \right)$;
   **end**
**end**

---

As noted in Section 2.3, feature maps $\phi_i$ can be ones from well-known kernels or problem-specific functions. If we use the concatenation as $\text{COMBINE}_i$ in Algorithm 1, because $\|f^i(v) - f^i(v')\|^2 = \|f^{i-1}(v) - f^{i-1}(v')\|^2 + \|\hat{f}^i(v) - \hat{f}^i(v')\|^2$, both the similarities between $f^{i-1}$ and between the sets of neighbors' attributes are incorporated into $f^i(v)$. The steps in a single iteration correspond to *transform-sum-cat* in this case.

After the node-wise update, we keep the set of updated attributes and discard the adjacency. To compare two graphs $G$ and $G'$, we measure the distance between $\{f^k(v) : v \in V(G)\}$ and $\{f^k(v') : v' \in V(G')\}$ using another kernel of choice. An example is to use the Gaussian kernel between the sum of node attributes.

**Extensions.** If the adjacency is not represented as a binary value but instead a number $w_{uv}$ is assigned to the edge from $u$ to $v$, we may reflect the weights by replacing the set representation $\sum_{u \in \mathcal{N}(v)} g^i(u)$ with $\sum_{u \in \mathcal{N}(v)} w_{uv} \cdot g^i(u)$; see Section 2.3. If an edge $uv$ has an attribute $e(u, v)$ then instead of set $\{f^i(u) : u \in \mathcal{N}(v)\}$ we consider set $\{\text{COMBINE}_i^e \left( e(u, v), f^i(u) \right) : u \in \mathcal{N}(v)\}$

with an edge-combination function $\text{COMBINE}_i^e$. That is, we are interested in similarity not merely between the sets of node attributes, but between the sets of pairs of edge attributes and node attributes.

In practice, the graph size and the node degrees may be too large to handle. We may choose to sample some of the neighbor attributes, to apply transform-mean then to multiply the node degree, which simulates the transform-sum operation.

## 3.1 WLS kernel

To test our idea, we implemented a WLS-based graph kernel. We report its classification accuracy on the TU datasets (Kersting *et al.*, 2016) in Section 5.1. Here, we describe which components are built into our kernel.

Algorithm 1 requires the iteration number $k$ and feature maps $\phi_i$ for all iterations. We set $k = 5$. For the feature map $\phi_i$, we chose the second-order Taylor approximation of the Gaussian kernel $K(x, y) = \exp\left(-\frac{\|x-y\|^2}{2}\right)$ so that $\phi(x)$ has entries 1, $x_i$, and $x_i x_j / \sqrt{2}$ all multiplied by $\exp(-\|x\|^2/2)$. See Appendix B.1 for the derivation. For functions $\text{COMBINE}_i$ we tested two options: (1) the sum $\phi(f^{i-1}(v)) + \hat{f}^i(v)$ and (2) the concatenation $f^{i-1}(v) \oplus \hat{f}^i(v)$.

Once the node-wise update is done, as noted above, we discard the adjacency and consider the set of updated node attributes. In Section 5.1, we used the Gaussian kernel between the sum of the final node attributes to compute the graph similarity.

**Dimensionality reduction.**     As we apply the feature maps iteratively, the dimension quickly surpasses the memory limit. For example, PROTEINS_full from the TU datasets (Kersting *et al.*, 2016) contains node attributes in $\mathbb{R}^{29}$. As a result, after three iterations with the above $\phi_i$, it yields more than five billion numbers for a single node. We must reduce the dimension.

What properties must our dimensionality reduction have? Recall from Section 2.3 that the set distance between $X$ and $Y$ is computed as $\|\sum_{x \in X} \phi(x) - \sum_{y \in Y} \phi(y)\|$, the Euclidean norm of the difference between set representations. Therefore, we would like to preserve the Euclidean norm through the reduction as much as possible. Motivated by the Johnson–Lindenstrauss lemma (Johnson & Lindenstrauss, 1984), we multiply $\text{COMBINE}_i\left(f^{i-1}(v), \hat{f}^i(v)\right)$ of high dimension $D$ with a random $d \times D$ matrix, say $M_i$. The entries of $M_i$ are i.i.d. random samples from the normal distribution, and the column norms of $M_i$ are normalized to 1. We apply this $M_i$ to all nodes at the $i$-th iteration. For a detailed explanation and empirical tests of stability, see Appendix C.2. We set $d = 200$; hence, after each WL-iteration, all nodes have a vector of dimension 200.

## 3.2 WLS neural network

Now, we propose a graph neural network based on Weisfeiler–Leman similarity. The node attributes are updated using Algorithm 1. We set $k = 4$ to compare the performance with other GNN models from Dwivedi *et al.* (2020). Each transformation $\phi_i$ is a three-layer multi-layer perceptron (MLP), where each layer consists of a linear transformation, 1D batch normalization, ReLU, and dropout in sequence. If the option *residual* is turned on, we add the input to the output of MLP after linearly transforming the input to match the dimension. All hyperparameters are listed in Appendix E.

After applying $\phi_i$ by going through the corresponding MLP, we use another 1D batch normalization. Then, for each node, we sum up the outputs of neighbors to obtain representation $\hat{f}^i(v)$ of the neighborhood. Further, we use $\text{COMBINE}_i\left(f^{i-1}(v), \hat{f}^i(v)\right) = \phi_i(f^{i-1}(v)) \oplus \hat{f}^i(v)$.

Again, following the experimental setup of Dwivedi *et al.* (2020), we place an MLP classifier with two hidden layers on top of the output vectors of Algorithm 1 for node classification. For graph classification, the averages of all node representations are put into the same classifier. We do not use the edge attributes, even if given, for these two tasks.

In the graph regression experiment, we additionally test an extended model that uses the edge attributes. The graphs in ZINC dataset has categorical edge attributes from $\{0, 1, 2\}$. Thus we assign each WL-iteration with its own learnable edge embeddings for the attributes. Let us denote $t_{uv}$ the attribute of edge $uv$ and $e_i(t_{uv})$ the corresponding embedding for the $i$-th iteration. Instead of

Table 1: Graph classification results on the TU datasets via WLS kernels with different aggregations. The numbers are mean test accuracies over ten splits. Bold-faced numbers are the top scores for the corresponding datasets. The proposed aggregation (WLS) shows strong performance compared with other aggregations from the literature. See Section 5.1.

| Aggregation | BZR | COX2 | DHFR | ENZYMES | PROTEINS | Synthie |
|---|---|---|---|---|---|---|
| GAT | 83.21±4.52 | **79.26±3.54** | 67.74±4.00 | 58.83±6.85 | 74.02±5.72 | 46.00±3.69 |
| GCN | 80.49±3.22 | 78.60±1.52 | 67.74±4.91 | 60.67±7.98 | 73.04±4.70 | 45.72±3.72 |
| GraphSAGE | 77.53±3.73 | 79.01±2.42 | 67.61±3.48 | 58.83±6.85 | 74.47±5.59 | 46.00±3.69 |
| WWL | 79.02±2.04 | 78.79±1.27 | 67.49±6.05 | 58.83±6.85 | 73.04±4.70 | 46.00±3.69 |
| WLSLin | 79.99±3.04 | 77.93±2.42 | 68.26±2.59 | 58.83±6.85 | 74.39±3.14 | 55.00±7.22 |
| WLS | **83.45±6.49** | 77.95±3.48 | **77.92±4.78** | **68.00±3.99** | **75.38±4.26** | **86.79±5.82** |

$\sum_{u \in \mathcal{N}(v)} \phi_i(f^{i-1}(u))$ as the neighborhood representation for node $v$, we use $\sum_{u \in \mathcal{N}(v)} \phi_i(e_i(t_{uv}) \oplus f^{i-1}(u))$. The rest are the same as in the WLS classification network.

# 4 Related work

**Graph kernels** Most of the graph kernels inspired by the Weisfeiler–Leman test act only on graphs with discrete (categorical) attributes. Morris *et al.* (2016) extend discrete WL kernels to continuous attributes; however, its use of hashing functions cannot reflect the continuous change in attributes smoothly. Propagation kernel (Neumann *et al.*, 2016) is another instance of hashing continuous attributes, which shares the same problem. WWL (Togninalli *et al.*, 2019) is a smooth kernel; however, the Wasserstein distance at its core makes it difficult to scale.

The kernels based on matching or random walks (Feragen *et al.*, 2013; Orsini *et al.*, 2015; Kashima *et al.*, 2003) are better suited for continuous attributes. Their speed can be drastically increased with explicit feature maps (Kriege *et al.*, 2019). However their construction often requires large auxiliary graphs, resulting again in scalability issues.

The deep graph kernels Yanardag & Vishwanathan (2015) are interesting approach to learn the similarity using Noise Contrastive Estimation Gutmann & Hyvärinen (2010). They collect substructures with specific co-occurence definitions and train their embeddings analogously to Word2Vec Mikolov *et al.* (2013). The pre-training stage and slow sampling of substructures makes its use cases different from our focus.

In comparison with fast kernels, the WLS kernel smoothly reflects the continuous change in attributes. Its simple structure combined with locality makes it easy-to-scale and easy-to-implement with existing deep learning frameworks. For further details on graph kernels, we redirect to two excellent surveys (Kriege *et al.*, 2020; Nikolentzos *et al.*, 2019).

**Graph neural networks.** The proposed GNN fits into the MPNN (Gilmer *et al.*, 2017) framework. Many MPNN-type GNNs can be analyzed upon the WL-framework, as shown by Xu *et al.* (2019). Transform-sum-cat is an instance of the WL-framework where transform-sum corresponds to AGGREGATE and cat to COMBINE. The notable differences of WLS-GNN from GIN and other popular GNN models are the order of operations and the usage of concatenation to distinguish the central node from its neighbors.

Although one of the earliest studies (Gori *et al.*, 2005; Scarselli *et al.*, 2009) applied transformation before aggregation, many modern GNNs follow the aggregation-first scheme, as noted by Xu *et al.* (2019). Furthermore, the theoretical analyses of such GNNs (Dehmamy *et al.*, 2019; Magner *et al.*, 2020; Morris *et al.*, 2019) focus on the distinguishing power of GNNs. However, we lack a discussion from the perspective of how similar the representations from different graphs are. Xu *et al.* (2019) called for future works extending their analysis to the continuous regime. This article can be a step toward an answer.

Table 2: Node classification results for Stochastic Block Model (SBM) datasets. The test accuracy and training time are averaged across four runs with random seeds 1, 10, 100, and 1000. WLS obtains the highest accuracy and is close to the best speed. See Section 5.2.

| Model | SBM_PATTERN | | SBM_CLUSTER | | # Parameters |
|---|---|---|---|---|---|
| | Accuracy | Time (h) | Accuracy | Time (h) | |
| GAT | 95.63±0.26 | 12.9 | 59.17±0.52 | 7.5 | 109936 |
| GatedGCN | 98.89±0.07 | 5.9 | 60.20±0.48 | 4.3 | 104003 |
| GCN | 80.00±1.01 | 4.5 | 55.23±1.07 | 2.3 | 100923 |
| GIN | 99.03±0.04 | 0.5 | 58.55±0.17 | 0.4 | 100884 |
| GraphSage | 89.34±1.49 | 5.1 | 58.36±0.18 | 2.5 | 98607 |
| MoNet | 98.94±0.05 | 29.5 | 58.42±0.36 | 19.0 | 103775 |
| WLS(ours) | **99.08±0.02** | 0.6 | **60.49±0.41** | 0.5 | 78452 |

# 5 Experiments

We present three sets of experiments to show the efficacy of our proposal. In Section 5.1, we test the transform-sum-cat against several aggregation operations from GNN literature, comparing their performances in graph classification. In Section 5.2, we show that a simple GNN based on transform-sum-cat can outperform popular GNN models in node classification and graph regression. In Section 5.3, we present a successful application of WLS in adversarial learning of graph generation with enhanced stability.

In all experiments, except for graph generation, we use the experimental protocols from the benchmarking framework[1] (Dwivedi *et al.* , 2020). For a fair comparison, the benchmark includes the datasets with fixed splits as well as reference implementations of popular GNN models, including GAT (Veličković *et al.* , 2018), GatedGCN (Bresson & Laurent, 2017), GCN (Kipf & Welling, 2017), GIN (Xu *et al.* , 2019), GraphSAGE (Hamilton *et al.* , 2017), and MoNet (Monti *et al.* , 2017).

## 5.1 Graph classification via graph kernels

In this subsection, we test the proposed *transform-sum-cat* against other neighborhood aggregation methods from GNN research. All models in Table 1 have the same structure and experimental settings as the WLS kernel (Section 3.1), except for the neighborhood aggregation. For comparison, we chose simple averaging of GraphSAGE (Hamilton *et al.* , 2017), degree-normalized averaging of GCN (Kipf & Welling, 2017), attention-based weighted averaging of GAT (Veličković *et al.* , 2018), and a customized aggregation from WWL (Togninalli *et al.* , 2019). To isolate the effect of transformation, we also report the WLS kernel performance with the identity feature map as WLSLin (*sum-cat* instead of *transform-sum-cat*). We input the obtained graph representations into the C-regularized Support Vector Classifier with Gaussian kernel in Scikit-learn (Pedregosa *et al.* , 2011).

Table 1 reports mean test accuracies across 10 splits given by the benchmarking framework, where the hyperparameters (including the number of iterations) are chosen based on the mean validation accuracy across the splits. Appendix E.1 lists the details of aggregations and hyperparameter ranges. On all datasets except for COX2, the WLS kernel outperforms other aggregations. Due to the stochasticity introduced by the dimensionality reduction in the WLS kernel, we run WLS experiments four times with random seeds 1, 10, 100, and 1000. The result showing the stability of the WLS kernel is in Appendix D.2.

We remark that the objective of this experiment is to show the relative strength of transform-sum-cat as an aggregation method without parameter learning, but not to show the strength of WLS kernel as a graph classification model. In fact, the classification performance by graph neural networks, Table 3, shows better accuracy on the same datasets.

Table 3: Graph classification on TU datasets via graph neural networks. ENZ. for ENZYMES, PRO. for PROTEINS_full, Synth. for Synthie. The numbers in the second sets of columns are mean test accuracies over ten splits, averaged over four runs with random seeds 1, 10, 100, and 1000. MRR stands for Mean Reciprocal Rank and Time indicates the accumulated time for single run across all six datasets. Bold-faced numbers indicate the best score for each column. See Section 5.2.

| | Model | BZR | COX2 | DHFR | ENZ. | PRO. | Synth. | MRR | Time (hr) | #Params |
|---|---|---|---|---|---|---|---|---|---|---|
| Isotropic | GCN | 84.63 | 77.08 | 76.06 | 62.38 | 75.24 | 93.03 | 0.200 | 4.9 | 78019 |
| | GIN | 83.49 | 79.01 | 76.62 | 65.67 | 65.58 | 93.05 | 0.202 | 4.0 | 77875 |
| | GraphSage | 81.61 | 76.89 | 74.80 | 68.58 | 75.85 | 97.70 | 0.291 | 5.2 | 81171 |
| | MLP | 82.98 | 76.82 | 73.01 | 54.88 | 74.37 | 48.43 | 0.135 | 1.2 | 60483 |
| | WLS(ours) | 84.64 | 79.07 | 76.95 | **70.42** | 63.07 | **98.87** | **0.576** | 3.9 | 64751 |
| Anisotropic | GAT | **85.31** | 78.52 | 76.39 | 66.63 | 75.62 | 95.08 | 0.358 | 38.4 | 78531 |
| | GatedGCN | 84.39 | **80.21** | 76.86 | 68.92 | 75.65 | 96.31 | 0.458 | 12.2 | 87395 |
| | MoNet | 83.64 | 79.64 | **78.22** | 57.63 | **76.92** | 91.46 | 0.498 | 10.3 | 82275 |

## 5.2 Graph neural network

In this subsection, we report the performance of the proposed GNN model (WLS) based on *transform-sum-cat* (Section 3.2). We used the implementation and hyperparameters for all models other than WLS as provided by the benchmarking framework[2] (Dwivedi *et al.* , 2020). All reported numbers are averaged across four different runs with random seeds 1, 10, 100, and 1000. The details of the experiments are in Appendix E.2.

**Node classification.** The benchmarking framework provides two community detection datasets for node classification: SBM_PATTERN and SBM_CLUSTER. Both are generated by the stochastic block model (SBM); see Abbe (2018). Graphs in SBM_PATTERN have six communities, where the task is binary classification: separating one specific community from the others. The communities differ by internal and external edge density. The node attributes are randomly assigned from $\{0, 1, 2\}$. Graphs in SBM_CLUSTER also have six communities, and the task is to classify each node as belonging to one of the six communities. Only one node for each community is assigned a community index from 1 to 6 as the node attribute, and all other nodes have attribute 0.

Table 2 shows the test accuracies and training times of all models on SBM_PATTERN and SBM_CLUSTER. WLS obtains the highest accuracy on both, being the second-fastest model.

**Graph classification.** As in graph kernel experiment, we test our WLS-based neural network on classification task of TU datasets BZR, COX2, DHFR, ENZYMES, PROTEINS_full and Synthie. The reported numbers in Table 3 are the average of mean test accuracy across the ten splits given by Dwivedi *et al.* (2020), over four different random seeds 1, 10, 100 and 1000.

Although the variances are quite large (c.f. Appendix D.3), WLS reports the highest mean accuracy on all datasets but PROTEINS among isotropic GNNs, while beating anisotropic ones on two datasets. As a measure of overall rankings we also report the Mean Reciprocal Rank for which WLS achieves the highest score.

**Graph regression.** The benchmarking framework provides one dataset extracted from ZINC (Sterling & Irwin, 2015) for graph regression. The graphs in ZINC are small molecules, whose node attributes are atomic numbers, and edge attributes are one of three bond types. The target labels are the constrained solubility (Jin *et al.* , 2018). Table 4 reports the mean absolute error (MAE) between the label and the predicted number for the provided test set. WLS already significantly outperforms the other models, and WLS-E, an extension using the edge attributes, lowers the error further.

## 5.3 Neural graph generation

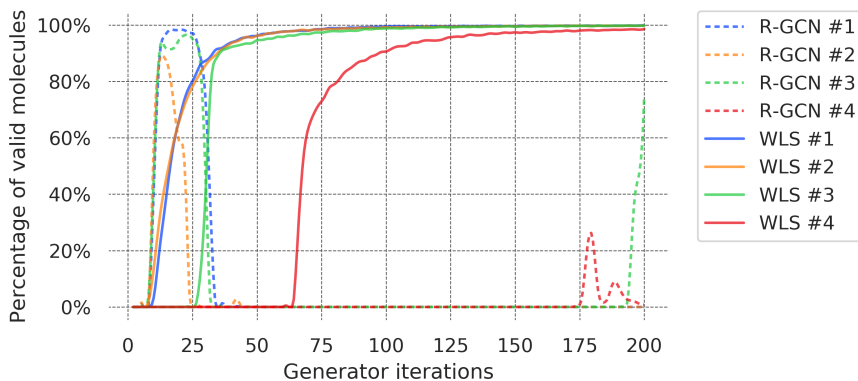

Figure 3: Percentage of valid molecules ($y$-axis) out of 13,317 generated molecules, plotted over the number of generator iterations ($x$-axis). Dashed curves correspond to 4 different runs with the original R-GCN-based discriminator. Solid curves correspond to the WLS discriminator. While R-GCN discriminator collapses to 0 valid molecules after approximately 20 mostly-valid generations, the generations from the WLS discriminator continue to be valid. See Section 5.3.

Generating molecules is the most popular application of neural graph generation because of its huge economic value. However, previous works generating a whole molecular graph at once (De Cao & Kipf, 2018; Simonovsky & Komodakis, 2018) suffer from low chemical validity of generated molecules. Hence, most studies either generate a string representation or devise iterative generation methods; c.f. Elton *et al.* (2019). Here, we present a case where a simple adoption of WLS greatly enhances the stability of one-shot generation.

MolGAN (De Cao & Kipf, 2018) uses WGAN-GP (Gulrajani *et al.* , 2017) framework to generate small molecules. We started from a re-implementation[3] of MolGAN in PyTorch, replacing the original discriminator based on R-GCN (Schlichtkrull *et al.* , 2018) with the WLS discriminator. Figure 3 reports the percentage of valid molecules out of 13,317 generated ones after each generator iteration. For validity tests, we used RDKit (Landrum, 2019) function SanitizeMol. According to our results, the generation validity with the WLS discriminator is much more stable than that with the R-GCN based discriminator. See Appendix E.3 for details.

Table 4: Graph regression on ZINC dataset via graph neural networks. The numbers are the mean absolute error (MAE) on the test set of 1,000 molecules. The suffix "-E" indicates that the model uses edge attributes. See Section 5.2.

| Model | MAE |
|---|---|
| GAT | $0.462\pm0.010$ |
| GatedGCN-E | $0.362\pm0.001$ |
| GCN | $0.469\pm0.014$ |
| GIN | $0.429\pm0.036$ |
| GraphSAGE | $0.422\pm0.006$ |
| MoNet | $0.416\pm0.014$ |
| WLS(ours) | $0.332\pm0.007$ |
| WLS-E(ours) | $\mathbf{0.315\pm0.003}$ |

## 6   Conclusion

Deep learning on graphs naturally calls for the study of graphs with continuous attributes. Previous analyses of GNNs identified the cases when non-identical graphs had the same learned representations. However, it has been unclear how similarities between input graphs could be reflected in the distance between GNN representations. Moreover, in the field of graph kernels, a dichotomy exists. On the one hand, we have fast and efficient kernels, which cannot reflect a smooth change in the node attributes. On the other hand, we have smooth matching-based kernels, which are slow and costly.

In this paper, we present an approach that reflects the similarity in GNN architecture. The resulting simple GNN model shows strong empirical performance and efficiency. Using the same idea, we introduce a graph kernel that smoothly reflects a continuous change in attributes, while also being simple and fast. We believe that this study demonstrates that starting from the perspective of continuity can help to improve GNN architectures.

## Acknowledgment

We appreciate Seok Hyun Hong, You Young Song, and Jiung Lee for their reviews of an early draft. The critique from Hong particularly helped enhancing the readability. We thank Eunsoo Shim, Young Sang Choi, and Hoshik Lee for setting up the environment for independent research in Samsung Advanced Institute of Technology. We thank all anonymous reviewers for their constructive comments.

## Broader impact

This article mainly discusses two topics: how to measure similarity between graphs, and how to learn from graphs. One of the most important subjects in both fields is the molecular graph. A chemically meaningful similarity between molecules helps find new drugs and invent new materials of great value. Many chemical search engines support similarity search based on fingerprints, which indicate the existence of certain substructures. The fingerprints have been useful to find molecules of interest, but they are inherently limited to local properties. The proposed graph similarity is simple, fast and efficient. The proposed graph neural network reports particular strength in molecular property prediction and molecular graph generation, albeit not studied extensively. It is possible that the proposed algorithms provide another, global perspective to molecular similarity.

Another task for which the proposed neural network showed strength is the node classification. The node classification is mostly used to automatically categorize articles, devices, people, and other entities in interconnected networks at large scale. Some related examples include identifying false accounts in social network services, classifying a person for a recommendation system based on its friends' interest, and detecting malicious edge-devices in Internet of Things or mobile networks. As with every machine learning applications, assessing and understanding the data is crucial in such cases. Especially in graph-structured data, we believe that the characteristic of data is the most important factor in deciding which graph learning algorithm to use. It is necessary to understand the principle and limitation of an algorithm to prevent failure. For example, our method has two caveats. First, it uses sum to collect information from the neighbors, and hence more suitable when the counts indeed matter and not just the distributions. Second, our method decides the similarity between two graphs using the local information. Hence when the "global" graph properties such as hamiltonicity, treewidth, and chromatic number are the deciding factor, our algorithm might not be the best choice.

Graph learning in general are being applied to more and more tasks and applications. Some of the examples include recommendation systems, transportation analysis, and credit assignments. However, the study of risks regarding graph learning, such as adversarial attack, privacy protection, ethics and biases are still at an early stage. In practice, we should be warned about such risks and devise testing and monitoring framework from the start to avoid undesirable outcomes.

## Footnotes

[1]The experiments in this paper are compared against Benchmark v1. However, the framework has been substantially updated after this paper was submitted.

[2]https://github.com/graphdeeplearning/benchmarking-gnns

[3]https://github.com/yongqyu/MolGAN-pytorch

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
