[Supplementary Material]

# Supplementary material for the paper
# "A Graph Similarity for Deep Learning"

## A   Datasets

Here, we describe all the datasets used in our paper with sources. The basic statistics is in Table 1.

**TU datasets**   The TU datasets are at `https://ls11-www.cs.tu-dortmund.de/staff/morris/graphkerneldatasets`. It is a collection of many graph classification datasets collected by Technische Universität Dortmund. Datasets may have graphs with categorical or continuous node attributes as well as edge attributes. As we focus on the graphs with continuous node attributes, we selected BZR, COX2, DHFR, ENZYMES, PROTEINS_full and Synthie, which have at most 2,000 graphs for fast test. The dataset AIDS is excluded since all models achieved close to or above 99% test accuracy on it.

For train-validation-test split we used the tool from the benchmarking framework[1] (Dwivedi *et al.* , 2020). The function  **get_all_split_idx**  in **data/TUs.py** builds and saves a random 10-split. The saved splits are included in our code.

**Node classification datasets**   The benchmarking framework (Dwivedi *et al.* , 2020) provided two community detection datasets, SBM_PATTERN and SBM_CLUSTER for node classification. To download, either see `https://github.com/graphdeeplearning/benchmarking-gnns/blob/master/docs/02_download_datasets.md` or get the pickle files directly from

- SBM_CLUSTER : `https://www.dropbox.com/s/edpjywwexztxann/SBM_CLUSTER.pkl`
- SBM_PATTERN : `https://www.dropbox.com/s/zf17n6x6s441s14/SBM_PATTERN.pkl`

The following description is re-phrasing of the explanation in Dwivedi *et al.* (2020).

A graph in either dataset have six communities in total, where a community is randomly generated from three parameters, number of nodes $n$, internal edge density $p$, and external edge density $q$. The density $p$ decides the probability of two nodes from the same community being adjacent, independently from other edges. The density $q$ is the edge probability between two nodes from different communities.

For SBM_PATTERN the task is to distinguish one special community $P$ from the others. The special community has $n = 20$, $p = 0.5$ and $q = 0.5$. Other communities have $n$ uniformly selected from $[5, 35]$, and set $p = 0.5$ and $q = 0.2$. The node attributes are randomly assigned from $\{0, 1, 2\}$. The nodes in $P$ have label 1 and other nodes have 0.

For SBM_CLUSTER the task is to identify which community each vertex belongs to. All communities have $n$ from $[5, 35]$, $p = 0.55$ and $q = 0.25$. Precisely one node from each community has informative node attribute from $\{1, 2, 3, 4, 5, 6\}$ which indicates its community. Other nodes have non-informative attribute 0.

**Graph regression dataset**   The benchmarking framework of Dwivedi *et al.* (2020) provided one dataset derived from ZINC (Sterling & Irwin, 2015) for graph regression. The graphs in ZINC are molecular graphs, whose nodes are atoms and edges are chemical bonds. All nodes and edges have categorical attributes. The node attributes correspond to the atomic numbers and re-labelled to $[0, 20]$. Edge attributes are from $\{1, 2, 3\}$ which correspond to single, double, and triple bond respectively.

The label for a graph in ZINC is constrained solubility used in Jin *et al.* (2018), which is given by $y(m) = \log P(m) - SA(m) - cycle(m)$, where $SA$ is the synthetic accessibility score and $cycle$ is the number of rings with more than six atoms.

Table 1: Statistics of graph datasets. Superscript $^*$ indicates the attributes are categorical variables converted to one-hot vectors. ZINC and QM9 are not used for classification and hence no graph classes.

| Dataset | #Graphs | #Classes | Avg.Nodes | Avg.Edges | Node dim | Edge dim |
|---|---|---|---|---|---|---|
| BZR | 405 | 2 | 35.75 | 38.36 | 3 | - |
| COX2 | 467 | 2 | 41.22 | 43.45 | 3 | - |
| DHFR | 467 | 2 | 42.43 | 44.54 | 3 | - |
| ENZYMES | 600 | 6 | 32.63 | 62.14 | 18 | - |
| PROTEINS_full | 1113 | 2 | 39.06 | 72.82 | 29 | - |
| Synthie | 400 | 4 | 95.00 | 172.93 | 15 | - |
| SBM_CLUSTER | 12000 | 6 | 117.20 | 4301.72 | $7^*$ | - |
| SBM_PATTERN | 14000 | 2 | 117.47 | 4749.15 | $3^*$ | - |
| ZINC | 12000 | - | 23.16 | 49.83 | $28^*$ | $4^*$ |
| QM9 | 133171 | - | 8.80 | 9.40 | $5^*$ | $5^*$ |

**Graph generation dataset**    We used a re-implementation[2] of MolGAN (De Cao & Kipf, 2018) in PyTorch (Paszke $et\ al.$ , 2017) for graph generation experiment. The data to learn is from QM9 (Ramakrishnan $et\ al.$ , 2014) dataset and prepared by the script **data/download_dataset.sh**.

# B    Feature maps

## B.1    Feature map for polynomial kernels and Gaussian kernels

We shall derive an explicit feature map for the polynomial kernel $K_P : \mathbb{R}^2 \times \mathbb{R}^2 \to \mathbb{R}$, $K_P(x, y) = (1 + \langle x, y \rangle)^2$ and the Gaussian kernel $K_G : \mathbb{R}^d \times \mathbb{R}^d \to \mathbb{R}$, $K_G(x, y) = \exp(-\|x - y\|^2/2)$. Generalizations to other dimensions of domain and other constants are easily obtained in a similar manner.

**Feature map for $K_P$.**    We shall find a mapping $\phi_P : \mathbb{R}^2 \to \mathbb{R}^D$ such that $K_P(x, y) = \langle \phi_P(x), \phi_P(y) \rangle$ for all $x, y \in \mathbb{R}^2$. We expand $K_P(x, y)$ using their coordinates $x = (x_1, x_2)$ and $y = (y_1, y_2)$.

$$
\begin{aligned}
K_P(x, y) &= (1 + \langle x, y \rangle)^2 \\
&= 1 + 2\langle x, y \rangle + \langle x, y \rangle^2 \\
&= 1 + 2x_1 y_1 + 2x_2 y_2 + x_1^2 y_1^2 + 2x_1 x_2 y_1 y_2 + y_1^2 y_2^2 \\
&= \left\langle \left(1, \sqrt{2}x_1, \sqrt{2}x_2, x_1^2, \sqrt{2}x_1 x_2, x_2^2\right), \left(1, \sqrt{2}y_1, \sqrt{2}y_2, y_1^2, \sqrt{2}y_1 y_2, y_2^2\right) \right\rangle
\end{aligned}
$$

Therefore, $\phi_P(x) = \left(1, \sqrt{2}x_1, \sqrt{2}x_2, x_1^2, \sqrt{2}x_1 x_2, x_2^2\right)$ is a feature map for the polynomial kernel $K_P$.

**Feature map for $K_G$.**    Similarly to the case of $K_P$, we shall manipulate the formula for $K_G$ to find a feature map. However, it results in a mapping into $\mathbb{R}^\infty$ and hence we shall use an approximation.

Note that $K_G(x, y) = \exp(\|x - y\|^2/2) = \exp(-\|x\|^2/2)\exp(\langle x, y \rangle)\exp(-\|y\|^2/2)$. We apply the second-order Taylor approximation $\exp z \approx 1 + z + z^2/2$ so that

$$
K_G(x, y) \approx \exp(-\|x\|^2/2)(1 + \langle x, y \rangle + \langle x, y \rangle^2/2)\exp(-\|y\|^2/2).
$$

The middle term can be represented as an inner product of two vectors in a similar manner to $K_P$. Therefore we obtain

$$
\tilde{\phi}_G(x) = \exp\left(-\frac{\|x\|^2}{2}\right) \cdot \left(1, x_1, x_2, \ldots, x_d, \frac{x_1 x_1}{\sqrt{2}}, \frac{x_1 x_2}{\sqrt{2}}, \ldots, \frac{x_1 x_d}{\sqrt{2}}, \frac{x_2 x_1}{\sqrt{2}}, \ldots, \frac{x_d x_d}{\sqrt{2}}\right).
$$

## B.2 How polynomial kernel locates the leaf nodes

Let $T$ be a graph on four nodes drawn on the Euclidean plane, with one node at $(0,0)$ connected to three other nodes on the unit circle. Let the 2D coordinates be assigned as node attributes. If we take either the sum or average of the outer vertices' attributes then the result is uninformative of the outer nodes' locations. Let us consider a weighted sum. The weight for the outer node $(\cos\theta, \sin\theta)$ can reasonably be assumed to be a function of $(0,0)$ and $(\cos\theta, \sin\theta)$. Without further assumptions on the data, it is plausible to assume that the function is rotationally invariant, for example the Euclidean distance. In that case, the resulting aggregation again outputs the same value regardless of the locations of the outer vertices, as long as they are all on the unit circle. That is, most GNN aggregation methods using aggregate-transform cannot distinguish such graphs.

Now we show that if we apply a proper transformation first to the outer node attributes, then transform-sum have full information of the outer nodes. Let $\phi : \mathbb{R}^2 \to \mathbb{R}^d$ be the feature map for polynomial kernel $(1 + \langle x, y \rangle)^3$ built as described above. Let $(x_i, y_i)$ for $i = 1, 2, 3$ be the coordinates (attributes) of the outer nodes, located not just on the unit circle, but arbitrarily. Among other entries, $\phi((x,y))$ has $x, x^2, x^3$ and $y, y^2, y^3$. Therefore, from the set representation $\sum_i \phi((x_i, y_i))$ we can infer $\sum_i x_i$, $\sum_i x_i^2$, and $\sum_i x_i^3$. It is easy to see that three real numbers $a, b, c$ are uniquely determined by $a + b + c$, $a^2 + b^2 + c^2$, and $a^3 + b^3 + c^3$. Therefore the result of transform-sum has all the information to retrieve $x_1, x_2$ and $x_3$.

## B.3 Transform-sum has orientation-invariance and monotonicity

First, we begin by defining *orientation-invariance* and *monotonicity*. Let $V_0 = \{v_i \in \mathbb{R}^d : i = 1, 2, \ldots, k\}$ be a multiset of vectors, and let $F$ be a representation function mapping a non-empty multiset of vectors in $\mathbb{R}^d$ to a vector in $\mathbb{R}^D$. For a function $T : \mathbb{R}^d \to \mathbb{R}^d$, we write $T(V_0) = \{T(v_i) : v_i \in V_0\}$. We assume that $F$ has an associated distance $D_F$ between two sets $V, V'$ as $D_F(V, V') = d_F(F(V), F(V'))$ for $d_F : \mathbb{R}^D \times \mathbb{R}^D \to \mathbb{R}$. Function (or *representation*) $F$ is *orientation-invariant* if it has the following two properties:

1. For a rotation $R$ in the orthogonal group $O(d)$, $D_F(V_0, R(V_0)) = D_F(V_0, R^t(V_0))$.

2. For a translation $T_\delta(v) = v + \delta$ for $\delta \in \mathbb{R}^d$, $D_F(V_0, T_{\delta_1}(V_0)) = D_F(V_0, T_{\delta_2}(V_0))$ if $||\delta_1|| = ||\delta_2||$.

The following definition is complex for rigor. Stating informally, we want that a small rigid transformation should strictly increase the dissimilarity as the transformation degree increases.

Representation $F$ is *monotone* if it has the following two properties:

1. Let $R$ from the orthogonal group $O(d)$ be a 2D-rotation with degree $\theta_0$. We write $R_\theta$ be the 2D-rotation in $O(d)$ with the same fixed axis and orientation as $R_0$ but with degree $\theta$ so that $R_{\theta_0} = R$. There exists $\theta > 0$ such that function $f(x) = D_F(V_0, R_x(V_0))$ is strictly increasing for $x \in [0, \theta]$.

2. Let $T_{\delta, \epsilon}(v) = v + \epsilon\delta$ be a translation with $\epsilon \in \mathbb{R}$ and $\delta \in \mathbb{R}^d$. For each $\delta$, there exists $\epsilon > 0$ such that the function $f(x) = D_F(V_0, T_{\delta, x}(V_0))$ is strictly increasing for $x \in [0, \epsilon]$.

Let us fix $d_F$ as the euclidean distance. For the Gaussian kernel with infinite-dimensional feature map, we consider the $l^2$ space with $d_F$ being the canonical distance. For a kernel $K$ on $\mathbb{R}^d$, let $F_K$ be the representation obtained by applying a feature map of $K$ then sum up the results.

We show the following.

1. Let $K$ be a kernel which can be expressed as $K(v, w) = \sum_{i=0}^{\infty} c_i \langle v, w \rangle^i$ for some $c_i \in \mathbb{R}$. Then $F_K$ has orientation-invariance (1).

2. Let $K$ be a kernel which can be expressed as $K(v, w) = \sum_{i=0}^{\infty} c_i ||v - w||^i$ for some $c_i \in \mathbb{R}$. Then $F_K$ has orientation-invariance (2).

3. Let $K$ be a kernel which can be expressed as $K(v, w) = \sum_{i=0}^{\infty} c_i \langle v, w \rangle^i$ where $c_i \geq 0$ for all $i$ and there exists $c_i > 0$ for some $i \geq N$. Then for $V$ with $|V| \leq N$, $F_K$ is monotone.

4. If $K$ is the Gaussian kernel, then $F_K$ is monotone.

Note that most fast graph kernels start with discretizing the continuous attributes. Therefore, they are not orientation-invariant nor monotone. Also, aggregation methods from popular GNN models, such as mean-pool, sum-pool, or attention-based aggregations, are not monotone. They satisfy orientation-invariance (1) only trivially, by having $D_F = 0$ regardless of rotations.

Now we begin the proof.

First, we prove Statement 1. Note that

$$D^2_{F_K}(V_0, R(V_0)) = \|F_K(V_0) - F_K(R(V_0))\|^2 = \sum_{i,j} K(v_i, v_j) + \sum_{i,j} K(Rv_i, Rv_j) - 2 \sum_{i,j} K(v_i, Rv_j).$$

Since

$$\sum_{i,j} K(Rv_i, Rv_j) = \sum_{i,j} \sum_k c_k \langle Rv_i, Rv_j \rangle^k = \sum_{i,j} \sum_k c_k \langle v_i, v_j \rangle^k = \sum_{i,j} K(v_i, v_j),$$

we need to show $\sum_{i,j} K(v_i, Rv_j) = \sum_{i,j} K(v_i, R^t v_j)$.

It is enough to show that $\sum_{i,j} \langle v_i, Rv_j \rangle^k = \sum_{i,j} \langle v_i, R^t v_j \rangle^k$ for all $k$. Let $A$ be the matrix whose columns are $v_i$. Let $f_k$ be the function which maps a matrix $M$ to the sum of $k$-th power of the entries of $M$. Since $f_k(M) = f_k(M^t)$, we have

$$\sum_{i,j} \langle v_i, Rv_j \rangle^k = f_k(A^t R A) = f_k((A^t R A)^t) = f_k(A^t R^t A) = \sum_{i,j} \langle v_i, R^t v_j \rangle^k.$$

Proving Statement 2 is trivial again by the definition of $D^2_{F_K}$. For Statement 3, note that the function $f(x)$ in both properties of monotone representation are analytic up to $x$. Since $D_F$ is non-negative and $f(0) = 0$, we only need to show that $f(x)$ is not constantly zero. Following the argument in B.2, the map $F_K$ is injective and hence $f(x)$ is not constantly zero.

Now we prove Statement 4. We apply Statement 2 to the Taylor expansion of the Gaussian kernel to obtain the orientation-invariance (2). For orientation-invariance (1), we compare

$$D^2_{F_K}(V_0, R(V_0)) = \sum_{i,j} K(v_i, v_j) + \sum_{i,j} K(Rv_i, Rv_j) - 2 \sum_{i,j} K(v_i, Rv_j)$$

against $D^2_{F_K}(V_0, R^t(V_0))$. Since $\|v_i - v_j\| = \|Rv_i - Rv_j\|$, the first two summands cancel out in comparison between $D^2_{F_K}(V_0, R(V_0))$ and $D^2_{F_K}(V_0, R^t(V_0))$. We need $\sum_{i,j} \exp(-\rho \|v_i - Rv_j\|^2) = \sum_{i,j} \exp(-\rho \|v_i - R^t v_j\|^2)$. We expand $\exp$ using the Taylor expansion and substitute $\|v_i - Rv_j\|^2 = \|v_i\|^2 + \|v_j\|^2 - 2\langle v_i, Rv_j \rangle$. Since the expansion is absolutely convergent, we may rearrange the summands so that it is enough to show $\sum_{i,j} \langle v_i, Rv_j \rangle^k = \sum_{i,j} \langle v_i, R^t v_j \rangle^k$ for all $k$. The last statement is shown in Statement 1, finishing the proof.

We remark that the condition $|V| \leq N$ is necessary. Let $K$ be a polynomial kernel of degree less than $n$. For $v_i = (r \cos(\frac{2i}{n}), r \sin(\frac{2i}{n}))$ on $\mathbb{R}^2$ where $R_\theta$ is the 2D rotation of degree $\theta$ about the origin, then $D_K(V_0, R_\theta(V_0))$ is 0 regardless of $\theta$.

## C   Dimensionality reduction

### C.1   Random projection preserves the norm

Here, we explain the motivation to use random projection as a dimension reduction preserving the Euclidean norm. We shall first state the Johnson-Lindenstrauss Lemma.

**Lemma 1 (Johnson & Lindenstrauss (1984))** *Let $\epsilon \in (0, 1/2)$. Let $Q \subset \mathbb{R}^d$ be a set of $n$ points and $k = \frac{20 \log n}{\epsilon^2}$. There exists a Lipschitz mapping $f : \mathbb{R}^d \to \mathbb{R}^k$ such that for all $u, v$ in $Q$:*

$$(1 - \epsilon)\|u - v\|^2 \leq \|f(u) - f(v)\|^2 \leq (1 + \epsilon)\|u - v\|^2$$

The core idea of Johnson-Lindenstrauss Lemma is that if $A$ is a random $k \times d$ matrix whose entries are i.i.d. samples from the normal distribution, then $\|\frac{1}{\sqrt{k}} Ax\|$ approximates $\|x\|$ for $x \in \mathbb{R}^d$ within

(a) Varying the input data dimension. The approximation dimension is set to 200.

(b) Varying the approximation dimension. The input dimension is set to 200.

(c) Varying the scale of input data.

Figure 1: Testing the stability of proposed dimensionality reduction. Column-wise normalization to norm 1 is more stable than dividing each column by the constant $\sqrt{d}$ where $d$ is the target dimension of approximation. See Section C.2.

small relative error with high probability. Achlioptas (2003) proved similar results when $A$ is a random matrix with entries from $\{-1, 1\}$ or $\{-1, 0, 1\}$. Note that the target dimension $k$ in this random projection does not depend on neither $d$ nor $n$. Therefore, we empirically investigated how the random projection works in our case. See the next section for experiments.

## C.2  Experimental study of stability of random projections

The proof of Johnson-Lindenstrauss lemma in Achlioptas (2003) uses a $d \times D$ matrix $M$ to map a vector $v \in R^D$ to $Mv \in R^d$ while preserving the norm $\|v\|$ within small relative error. The proposed construction of $M$ is to sample each entry independently from the normal distribution, and multiply the resulting matrix by $\frac{1}{\sqrt{d}}$ which 'normalizes' the columns. For high $d$ the norms of the columns of $M$ are highly concentrated at 1. However, in our experiments, setting the column norms precisely to 1 gained a better stability.

In Figure 1, we tested the relative error between the Gram matrix of Gaussian kernel and the Gram matrix obtained by our approximation, which is to apply the second-order Taylor approximation of Gaussian kernel then reduce the dimension by multiplying a random matrix. The confidence intervals are obtained from 30 trials. In each experiment, we choose 30 random vectors and hence two Gram matrices of size $30 \times 30$ are obtained. The entry-wise maximum relative errors have been recorded. The Gaussian kernel is $K(x, y) = \exp\left(\frac{\|x-y\|^2}{2}\right)$.

In Figure 1 (a), we fixed the target dimension to 200 and varied the input dimension from 10 to 300. In Figure 1 (b), we fixed the input dimension to 200 and varied the approximation (target) dimension from 10 to 300. Because we use the second-order Taylor approximation of the exponential function, large values in the input will harm the approximation. Thus in Figure 1 (c), we tested how the scale of the input data affects the relative error. We first divide the data by the maximum absolute value then multiplied all entries by a constant, ranging from 0.7 to 3.5. The results show that column-normalized matrix is more suitable for our task. We set the target dimension to 200 to balance between computational feasibility and approximation stability.

## D  Further experiments

### D.1  WLS kernel with mean aggregation

We report graph classification result with WLS kernels from different aggregations. The kernel experiment reported in the main article used the sum of final node attributes as graph representation. Table 2 shows the result when the mean is used instead of sum. The tendency is similar.

Note that all aggregation methods reported the same test accuracy on PROTEINS dataset and four methods reported the same number on Synthie. It is because we include the features from iteration #0, which uses the raw node attributes directly for the graph representation. Accuracies from further

Table 2: Graph classification results on TU datasets via WLS kernels with different aggregations. The graph representation is the **average** of node attributes after at most five WL iterations. The numbers are mean test accuracies over 10-fold 8:1:1 splits. Bold-faced numbers are the top scores for the corresponding datasets. The strange numbers for PROTEINS and Synthie are due to iteration #0 having the best validation score. See D.3.

| Model | BZR | COX2 | DHFR | ENZYMES | PROTEINS | Synthie |
|---|---|---|---|---|---|---|
| GAT | 81.72±5.13 | **79.46±3.11** | 65.49±2.57 | 60.67±5.78 | **77.44±4.09** | 49.06±5.51 |
| GCN | 78.02±3.93 | 78.79±1.64 | 66.67±3.23 | 61.33±6.47 | **77.44±4.09** | 49.06±5.51 |
| GraphSage | 81.25±1.90 | 78.59±0.96 | 63.63±3.89 | 58.17±6.11 | **77.44±4.09** | 49.06±5.51 |
| WWL | 79.00±3.40 | 78.16±0.61 | 64.44±4.01 | 62.17±3.60 | **77.44±4.09** | 49.06±5.51 |
| WLSLin | **82.22±5.01** | 77.95±0.75 | 63.75±4.57 | 62.17±3.60 | **77.44±4.09** | 53.01±6.94 |
| WLS(ours) | **82.22±7.42** | 75.37±4.32 | **77.00±4.83** | **68.00±4.50** | **77.44±4.09** | **86.56±6.47** |

Table 3: Graph classification results on TU datasets via WLS kernels with different random seeds. The numbers are mean test accuracies over ten splits. Bold-faced numbers are the top scores for the corresponding datasets. The proposed aggregation (WLS) consistently outperforms other aggregations.

| Model | Seed | BZR | COX2 | DHFR | ENZYMES | PROTEINS | Synthie |
|---|---|---|---|---|---|---|---|
| GAT | - | 83.21±4.52 | **79.26±3.54** | 67.74±4.00 | 58.83±6.85 | 74.02±5.72 | 46.00±3.69 |
| GCN | - | 80.49±3.22 | 78.60±1.52 | 67.74±4.91 | 60.67±7.98 | 73.04±4.70 | 45.72±3.72 |
| GraphSage | - | 77.53±3.73 | 79.01±2.42 | 67.61±3.48 | 58.83±6.85 | 74.47±5.59 | 46.00±3.69 |
| WWL | - | 79.02±2.04 | 78.79±1.27 | 67.49±6.05 | 58.83±6.85 | 73.04±4.70 | 46.00±3.69 |
| WLSLin | - | 79.99±3.04 | 77.93±2.42 | 68.26±2.59 | 58.83±6.85 | 74.39±3.14 | 55.00±7.22 |
| WLS | 1 | 83.45±6.49 | 77.95±3.48 | 77.92±4.78 | 67.83±4.09 | 75.38±4.26 | 86.79±5.82 |
| WLS | 10 | 85.16±5.15 | 77.54±5.07 | 77.66±4.52 | 68.00±4.50 | 75.46±3.47 | 86.80±5.49 |
| WLS | 100 | 83.44±4.72 | 77.53±3.46 | 78.18±4.44 | 68.67±5.87 | **75.55±3.80** | 88.55±5.18 |
| WLS | 1000 | **85.19±5.53** | 76.65±4.39 | **79.63±4.04** | **69.17±5.29** | 75.19±4.42 | **88.77±5.38** |

iterations differ between aggregation methods, but on PROTEIN_dataset, all models obtained best validation and test accuracy when the raw attributes are used.

## D.2 WLS kernel stability

The proposed WLS kernel requires random projection to keep computational feasibility. Due to the introduced stochasticity, we tested graph classification performance of WLS kernels with four random seeds, 1, 10, 100, and 1000. The result is shown in Table 3. Although the exact numbers vary, WLSs consistently outperform other aggregations. The result in main article corresponds to seed 1.

## D.3 Graph classification via graph neural networks

In Table 4 we show a summary of the graph classification results using graph neural networks. The reported scores are mean test accuracy across ten splits. The results from same experiment with standard deviation is in Table 5. Dwivedi *et al.* (2020) noted that the anisotropic GNNs (GAT, GatedGCN, MoNet) in general perform better than isotropic GNNs. One possible explanation is that anisotropic aggregation generalizes isotropic aggregation and may help learning further by focusing on important neighbors more than less relevant neighbors.

Our proposed WLS-based GNN model is in fact isotropic. However, WLS outperformed all other models including anisotropic ones in node classification and graph regression. In graph classification, WLS outperforms all isotropic models across datasets, and competitive with the best anisotropic

Table 4: Graph classification on TU datasets via graph neural networks. ENZ. for ENZYMES, PRO. for PROTEINS_full, Synth. for Synthie. The numbers in the second sets of columns are mean test accuracies over ten splits, averaged over four runs each with random seed 1, 10, 100, and 1000. MRR stands for mean reciprocal rank and Time indicates the accumulated time for single run across all six datasets. Bold-faced numbers indicate the best score for each column.

| | Model | BZR | COX2 | DHFR | ENZ. | PRO. | Synth. | MRR | Time (h) | #Params |
|---|---|---|---|---|---|---|---|---|---|---|
| Isotropic | GCN | 84.63 | 77.08 | 76.06 | 62.38 | 75.24 | 93.03 | 0.200 | 4.9 | 78019 |
| | GIN | 83.49 | 79.01 | 76.62 | 65.67 | 65.58 | 93.05 | 0.202 | 4.0 | 77875 |
| | GraphSage | 81.61 | 76.89 | 74.80 | 68.58 | 75.85 | 97.70 | 0.291 | 5.2 | 81171 |
| | WLS(ours) | 84.64 | 79.07 | 76.95 | **70.42** | 63.07 | **98.87** | **0.576** | 3.9 | 64751 |
| Anisotropic | GAT | **85.31** | 78.52 | 76.39 | 66.63 | 75.62 | 95.08 | 0.358 | 38.4 | 78531 |
| | GatedGCN | 84.39 | **80.21** | 76.86 | 68.92 | 75.65 | 96.31 | 0.458 | 12.2 | 87395 |
| | MoNet | 83.64 | 79.64 | **78.22** | 57.63 | **76.92** | 91.46 | 0.498 | 10.3 | 82275 |

Table 5: Graph classification on TU datasets via graph neural networks, with standard deviation. Bold-faced numbers indicate the best mean for each column.

| | Model | BZR | COX2 | DHFR | ENZYMES | PROTEINS | Synthie |
|---|---|---|---|---|---|---|---|
| Isotropic | GCN | 84.63±5.86 | 77.08±4.72 | 76.06±3.44 | 62.38±7.25 | 75.24±3.47 | 93.03±4.14 |
| | GIN | 83.49±6.01 | 79.01±5.26 | 76.62±3.62 | 65.67±7.71 | 65.58±9.38 | 93.05±5.87 |
| | GraphSage | 81.61±5.21 | 76.89±5.58 | 74.80±4.06 | 68.58±5.20 | 75.85±3.25 | 97.70±1.93 |
| | WLS(ours) | 84.64±4.85 | 79.07±5.91 | 76.95±4.26 | **70.42±5.70** | 63.07±3.72 | **98.87±1.74** |
| Anisotropic | GAT | **85.31±6.01** | 78.52±6.07 | 76.39±4.51 | 66.63±5.07 | 75.62±3.52 | 95.08±3.12 |
| | GatedGCN | 84.39±5.41 | **80.21±4.70** | 76.86±3.37 | 68.92±6.12 | 75.65±3.47 | 96.31±2.54 |
| | MoNet | 83.64±6.49 | 79.64±5.19 | **78.22±3.33** | 57.63±18.31 | **76.92±3.38** | 91.46±4.61 |

models except on PROTEINS_full dataset. A notable advantage of WLS over anisotropic models is the training speed. Without the need to calculate different weights depending on neighbors, WLS achieves the fastest speed in this task.

### D.4   More data from graph regression

In the main article, we reported only the mean absolute error (MAE) for the graph regression task. Table 6 shows more data, including the validation set loss, two WLS models we tested, together with training time and the number of parameters. The difference between WLS1 and WLS2 is in the use of residual connection. WLS1 does not use it, and WLS2 applies the residual connection. We reported WLS1 in the main article.

### D.5   Further information about graph generation experiment

We showed in the main article that WLS-discriminator may help generating molecular graphs stably. While the chemical validity is a necessity, a good molecule generator must generate a diversity of molecules meeting the required criteria. In Figure 2, we plot the number of unique molecules generated by the WLS experiment. The numbers are clipped to 300 at most to emphasize the tail behavior. We observed that the number of unique molecules tends to decrease over time. The problem may be attributed to *mode collapse*, a persistent problem across GAN-based generations (Lala *et al.* , 2018).

Table 6: Graph regression results on ZINC dataset. The performance metric is mean absolute error (MAE). Two WLS models, with and without residual connection, outperfrom all the other models.

| Model | MAE: validation | MAE: test | Time (m) | #Params |
|---|---|---|---|---|
| GAT | 0.433±0.007 | 0.462±0.010 | 33.6 | 102385 |
| GatedGCN | 0.310±0.004 | 0.362±0.002 | 11.7 | 105875 |
| GCN | 0.443±0.013 | 0.469±0.014 | 4.7 | 103077 |
| GIN | 0.400±0.025 | 0.429±0.036 | 4.5 | 103079 |
| GraphSAGE | 0.392±0.008 | 0.422±0.006 | 7.1 | 105031 |
| MoNet | 0.388±0.012 | 0.416±0.014 | 18.1 | 106002 |
| WLS-E1 | 0.309±0.008 | **0.315±0.003** | 13.9 | 117950 |
| WLS-E2 | 0.313±0.008 | 0.322±0.007 | 13.7 | 117950 |

Figure 2: Number of unique molecules generated by the molecular graph generation experiment with WLS discriminator. We observed a degree of mode collapse. We clipped the numbers larger than 300 to make the tail behavior more visible.

# E    Reproducibility

In this section, we describe the information about models and hyperparameters that are necessary to reproduce the experimental results. The experiments are done on a single machine with two Intel Xeon E5-2620 v3 CPUs, and four GeForce GTX TITAN X GPUs. Each GNN experiment used only one GPU.

For easier re-implementation of the models, we illustrate the steps of node representations of (a) WLS-kernel and (b) WLS-GNN in Figure 3. Since the second-order Taylor approximation of the Gaussian feature map degrades when the vector has large entries, for WLS-kernel, we scale down the results of transform-sum-cat by multiplying $1/\sqrt{d}$ where $d$ is the input dimension, 200.

## E.1    Graph kernel

The steps of graph kernel experiments are as follows:

1. the initial node attributes are normalized linearly to have mean 0 and standard deviation 1 at each component across all graphs in the whole dataset.

2. the designated iterations are processed. Output from every iteration is stored, including the initial attributes.

3. the final node attributes are summed (or averaged in Table 2) to form the graph representation, for each iteration number from 0 to 5.

|          |          |
|----------|----------|
| (a) WLS-kernel | (b) WLS-GNN |

Figure 3: Illustration of the node representation updates of (a): WLS-kernel, and (b): WLS-GNN used in the experiments. WLS-kernel applies the second-order Taylor approximation of feature map for the Gaussian kernel. Then sum the neighbor attributes and concatenate with the central node's attribute. The result is multiplied by a random projection matrix to reduce the dimension. WLS-GNN is simpler. Apply MLP, sum the neighbors, then concatenate.

4. the graph representations are put into the Gaussian kernel $\exp(-g\|x\|^2)$ with a hyperparameter $g$.

5. the Gram matrix is then put into C-regularized SVM classifier.

The following hyperparameters are tuned based on the mean validation accuracy across the 10 splits: the iteration number, the width $g$ of Gaussian kernel, the SVM regularization constant $C$. The iteration number is selected from 0 to 5. The numbers $g$ and $C$ are from $\{10^i : i = -6, -5, -4, \ldots, 2, 3\}$. Further tuned hyperparameters specific to individual aggregations are stated below. We report the average and the standard deviation of the test accuracies across the 10 splits.

The neighborhood aggregations we used are the following.

**GAT** GAT (Veličković *et al.* , 2018) uses attention-based weighting of the neighbors. Without learnable parameters, we take softmax over inner products with temperature to decide the weights. That is, $f^i(v) = \sum_{u \in \mathcal{N}(v)} a_T(f^{i-1}(u), f^{i-1}(v)) f^{i-1}(u)$ where

$$a(f(u), a(v)) = \frac{\exp(\langle f(u), f(v)\rangle / T)}{\sum_{u' \in \mathcal{N}(v)} \exp(\langle f(u'), f(v)\rangle / T)}.$$

With the option residual, $f^{i-1}(v)$ is added to $f^i(v)$ before the next WL-iteration. The temperature $T$ is selected from $\{1, 10, 100\}$. Thus in total $6 \times 10 \times 10 \times 2 \times 3 = 3600$ models are considered for each split.

**GraphSAGE** We used the simplest form of GraphSAGE-Mean from Hamilton *et al.* (2017). That is, $f^i(v) = \frac{1}{|\mathcal{N}(v)|} \sum_{u \in \mathcal{N}(v)} f^{i-1}(u)$. With the option *residual* we add $f^{i-1}(v)$ to $f^i(v)$ before the next WL-iteration. The number of hyperparameter combinations is 1200.

**GCN** The degree-normalized averaging of GCN (Kipf & Welling, 2017) is implemented here as $f^i(v) = \sum_{u \in \mathcal{N}(v)} \frac{1}{\sqrt{\deg(u)\deg(v)}} f^{i-1}(u)$. The function $\deg$ denotes the node degree in graph. With the option *residual* we modify the formula as $f^i(v) = \frac{1}{\deg v + 1} f^{i-1}(v) + \sum_{u \in \mathcal{N}(v)} \frac{1}{\sqrt{(\deg u + 1)(\deg v + 1)}} f^{i-1}(u)$. The number of hyperparameter combinations is 1200.

**WWL** WWL (Togninalli *et al.* , 2019) used the following customized aggregation. We used it as-is.

$$f^i(v) = \frac{1}{2}\left(f^{i-1}(v) + \frac{1}{|\mathcal{N}(v)|} \sum_{u \in \mathcal{N}(v)} f^{i-1}(u)\right)$$

Without further hyperparameters, we tune the iteration number, $g$, and $C$ from 600 choices.

**WLS**    As stated in the main article, we proposed the following aggregation for WLS kernel.

$$f^i(v) = \text{COMBINE}_i \left( f^{i-1}(v), \sum_{u \in \mathcal{N}(v)} \phi_i(f^{i-1}(u)) \right).$$

We used the second-order Taylor approximation of Gaussian kernel $\exp(-\|x - y\|^2/\sigma^2)$ as the feature map $\phi_i$ with $\sigma$ from $\{10^i : i = -1, 0, 1\}$. The combination function $\text{COMBINE}_i$ is either the concatenation $f^{i-1}(v) \oplus \sum_{u \in \mathcal{N}(v)} \phi_i(f^{i-1}(u))$ or the sum $\phi_i(f^{i-1}(v)) + \sum_{u \in \mathcal{N}(v)} \phi_i(f^{i-1}(u))$ followed by random projection $M_i$ onto $\mathbb{R}^{200}$. The number of hyperparameter configurations is 3600.

## E.2    Graph neural network

Here, we discuss the hyperparameters for WLS-based neural network only. The other GNN models use hyperparameter settings from Dwivedi *et al.* (2020)[3]. All hyperparameters for our method are manually tuned among several options. The learning rate and optimizer settings are from Dwivedi *et al.* (2020), which we list below.

The optimizer is Adam (Kingma & Ba, 2015) with $\beta_1 = 0.9$, $\beta_2 = 0.999$, and weight decay set to 0.0. The initial learning rate is set to $1e - 3$, and if the validation loss is not decreased for 5 epochs then the learning rate is halved. The train stops either if the learning rate goes below $1e - 5$ or if the number of epochs reaches 1000.

**Node classification.**    The types and ranges of hyperparameters we considered for the WLS node classification network is in Table 7. The number of iterations is set to 4 because other GNN models from Dwivedi *et al.* (2020) uses 4 altogether. The number of MLP layers is tested between 3 and 4 because we focused on the non-linearity of the transformation. After having done all the experiments for the main article, we observed that setting the number of layers 2 with larger dimension may improve the performance in some cases. The hidden dimension scale decides the hidden dimension of MLP by scale $\times \max(\text{input dimension}, \text{output dimension})$. The dimensions are first tested with 200 to see whether the training progresses properly, then set to 50 to match the number of parameters of other GNN models. Fortunately, the number 50 worked nicely and not tuned further.

Table 7: Types and ranges of the hyperparameters for WLS node classification network. Bold-face indicates that the option is selected for the final report. Superscript $^*$ indicates the option is excluded in the preliminary test on other dataset.

| Hyperparameter | Range |
|---|---:|
| # iterations | **4** |
| MLP input dimension | [**50**, 200] |
| MLP output dimension | [**50**, 200] |
| MLP # layers | [**3**, $4^*$] |
| MLP hidden dimension scale | [**2**, 3] |
| dropout rate | [**0.0**, 0.1, $0.2^*$, $0.5^*$] |
| residual | [True, **False**] |

**Graph classification.**    The hyperparameters of our WLS neural network for graph classification task is similar to that of node classification; see Table 8. The choices become simpler by excluding the preliminary numbers. The optimizer setting is different to reflect the settings from Dwivedi *et al.* (2020). We wait 25 epochs for improvement in validation loss, and training stops if the learning rate becomes lower than $1e - 6$.

**Graph regression.**    For graph regression, see Table 9. An important difference is that we need the embedding dimension for edge attributes, which are categorical variables. We tested six arbitrarily selected combinations from the ranges. The final selection was done considering both the number of parameters and performance.

Table 8: Types and ranges of the hyperparameters for WLS graph classification network. Bold-face indicates that the option is selected for the final report.

| Hyperparameter | Range |
|---|---:|
| # iterations | **4** |
| MLP input dimension | **50** |
| MLP output dimension | **50** |
| MLP # layers | **3** |
| MLP hidden dimension scale | **2** |
| dropout rate | [0.0, **0.1**] |
| residual | [**True**, False] |

Table 9: Types and ranges of the hyperparameters for WLS graph regression network. Bold-face indicates that the option is selected for the final report.

| Hyperparameter | Range |
|---|---:|
| # iterations | **4** |
| edge embedding dimension | **50** |
| MLP input dimension | [**20**, 30, 40, 50] |
| MLP output dimension | [40, **50**] |
| MLP # layers | [2, **3**] |
| MLP hidden dimension scale | [1.5, **2**] |
| dropout rate | [**0.0**, 0.1] |
| residual | [**True**, False] |

### E.3 Graph generation

We minimally changed the code from MolGAN-PyTorch[4]. Besides the discriminator, we suppressed the warning from RDKit (Landrum, 2019) and changed console outputs.

The generator outputs two tensors of shape $A \in \mathbb{R}^{9 \times 5}$ and $E \in \mathbb{R}^{9 \times 9 \times 5}$. The former corresponds to atom types: $A_{ij}$ represents the atom type out of five possibilities, C, N, O, F, and padding for empty atom. The padding atom is for molecules of less than 9 atoms. The latter tensor corresponds to bond types: $E_{ijk}$ represents the type of bond connecting $i$-th and $j$-th atoms in $A$. We have five bond types: single, double, triple, and quadruple bonds, with padding for pairs of atoms without bond. Quadruple bonds do not appear in the dataset. Both tensors are put into softmax over the types after generated by MLP.

Our WLS discriminator has a tiny amount of parameters. After embedding the atoms into $\mathbb{R}^{50}$ with learnable matrix we do not introduce further parameters except the last regression layer. The transformations are not applied, i.e. we use the linear kernel. A WL-iteration consists of the following steps: (1) for each non-empty edge type, we sum the neighbor attributes using the edge values as weights (2) we take mean over the non-empty edge types to update the self-attribute. Note that the step (2) is equivalent to summing up then divide by 4. After four WL-iterations, we take the mean node attributes as the graph representation. It shall go through a linear layer with 1-dimensional output. Its sigmoid value is the discriminator score.

Note that the linear kernel WLS discriminator is not much different from R-GCN with its parameters removed, except the first embedding layer. We have not tested whether the stability is from reduced number of parameters or specific architecture.

## Footnotes

[1]`https://github.com/graphdeeplearning/benchmarking-gnns`. We tested on Benchmark v1 before the 2020 June update, which changed SBM datasets, added a couple new models, and substantially changed the hyperparameter ranges.

[2] `https://github.com/yongqyu/MolGAN-pytorch`

[3]`https://github.com/graphdeeplearning/benchmarking-gnns`

[4]`https://github.com/yongqyu/MolGAN-pytorch`