[Reviews · NeurIPS 2020]

Review 1

Summary and Contributions: The paper proposes a novel graph neural network architecture which is motivated both by the Weisfeiler-Lehman isomorphism test and kernel theory. In more detail, the paper states that the natural representation of a neighborhood of a node is not a simple sum but a sum of features in the kernel space, because this yields a meaningful kernel/distance between sets. Accordingly, the paper suggests that the lth-layer representation of a node should not be the transformed sum of previous layer representations in the neighborhood, but rather the sum of transformed previous-layer representations. [It should also be noted that the transformation suggested in this paper is more involved than usual, using an MLP first and then a Taylor approximation of the feature space of an RBF kernel, which is in turn dimensionality-reduced via random projections to keep the dimensionaliy of the representation manageable.] (this characterization is imprecise; as the authors pointed out in their feedback, the Taylor approximation and dimensionality reduction is replaced with an MLP) After all this processing, the inner product/Euclidean distance between node representations can be interpreted as a node kernel/distance and the inner product/Euclidean distance between the sum of node representations forms a well-defined graph kernel/distance. This scheme is evaluated in node classification, graph classification, graph regression, and graph generation tasks, yielding performance gains in all cases, with particularly impressive gains in graph kernel classification.

Strengths: The paper has numerous strengths, the most striking of which are the clear motivation of each architectural component as well as the extensive experimental evaluation. The paper starts from the straightforward Weisfeiler-Lehman test, which is closely related to many graph neural network variants, and then extends it logically with kernel concepts to construct a well-behaving graph similarity/distance. Each additional complexity is properly motivated either from the kernel world or from the isomorphism testing world and yields an overall architecture that is fairly simple to understand and yet very expressive and powerful. With respect to experiments, the paper uses an established methodology and compares against strong baselines from the literature which represent the state-of-the-art well. Still, especially in terms of graph classification, the proposed method yields impressive gains in performance, sometimes tens of percentage points. It is also impressive that the evaluation covers such a wide range of tasks, giving a very comprehensive impression of the capabilities of the proposed model. Finally, I regard the paper as highly valuable to the community, both because it provides improvements to a model type that has gained widespread interest in recent years and because the proposed extensions are easy to incorporate, such that the paper has the potential for widespread adoption in the next years.

Weaknesses: The main weakness of the paper is, in my eyes, the motivation for the similarity interpretation between kernels. If one already has a well-performing graph neural net for a task at hand, it is not immediately obvious that an interpretation of the resulting vector space in terms of similarities/distances is necessary. Here, it may be valuable to point to work which assumes that distances in the representation space have meaning, e.g. adversarial attacks, counterfactuals, or few-shot learning. Furthermore, the focus of the paper could be improved. Currently, several paragraphs discuss the problems of aggregation-first schemes with a very specific example (2.4), while related work is covered only briefly (with graph neural networks being compressed to a single paragraph). The proposed architecture is only presented textually, whereas a flow diagram of a single layer in the architecture would help to appreciate the modules involved. By contrast, Algorithm 1 is currently too abstract to understand the specific implementation proposed by the authors (it can be kept, but does not, in itself, suffice to grasp the approach). Finally, I believe that some of the related work is not covered (more on that below).

Correctness: The theoretical claims are correct and well justified and the experimenta methodology follows standards in the field. The only problem I see is that the graph classification results are obtained using only graph kernel variations of the baselines, whereas the graph neural network results for comparison are in the appendix. It should be noted in the text that the graph kernel classification is _not_ superior to an (arguably more straightforward) classification via an output layer (at least that is what I gather from the appendix). This is, again, related to the issue of motivation: The paper currently makes the added value of a kernel interpretation not sufficiently clear.

Clarity: The paper is overall well-written and easy to follow. The only issues in terms of clarity are mentioned above (i.e. a flow chart for the weisfeiler-lehman layer could be added and it should be made explicit that graph kernel classification does not outperform standard graph neural network classification).

Relation to Prior Work: The relation to related work is characterized correctly, in principle. However, graph neural networks are represented too briefly to be informative if one is not already very familiar with the fields and some works on deep learning and graph kernels are missing. At least one work on learned graph kernels (e.g. Yanardag and Vishwanathan, 2015, doi: 10.1145/2783258.2783417) should be included.

Reproducibility: Yes

Additional Feedback: With regards to reproducibility, I believe that the provided documentation is outstanding and makes reproducibility straightforward. With regards to the broader impact statement, I believe that potentially hazardous applications are not sufficiently covered. For example, if one would apply the proposed method to infer a 'risk score' (in terms of credibility or bad credit) of a person from their social network, this could be ethically questionable. It is worthwhile, I think, to caution readers against naively applying the proposed method for such tasks. typos and style: * line 132: I believe, the top index of \hat f should be i, not i-1. * in line 153, the concatenation should probably be \phi(f^{i-1}(v)) \oplus \hat f^i(v), i.e. the first argument of the concat operator should be first plugged in \phi.


Review 2

Summary and Contributions: The paper proposes a transform-sum-cat scheme for graph representation learning, in contrast to the sum-transform scheme that is typically applied in GNNs. Here, authors first identify problematic aspects of sum-transform schemes when applied on graphs with continuous node attributes, and proceed to present solutions which are both applicable for graph kernels as well as GNNs (called WLS). Experiments are conducted on a diverse range of tasks, ranging from graph classification, node classification, graph regression and graph generation.

Strengths: The paper is well-written and easy to follow, and I certainly enjoyed reading it. Its main contribution lies in the fact that GNNs, which rely on a (linear) feature transformation, lose expressive power when operating on continuous input node features. As a result, the authors propose to non-linearly transform node features before aggregating them. This scheme is unified into a single algorithm which is applicable for both graph kernels and GNNs. Furthermore, the experimental evaluation looks quite convincing to me since the proposed approach is tested on a wide range of different tasks, a welcome contribution!

Weaknesses: * Overall, the paper feels quite incremental to me. Transforming node representations before aggregation is not a novel concept, and can be found in literature quite often. This should be at least discussed in a revised manuscript. In particular, the official formulation of GIN [1] is based on this concept, and only drops the first transformation in case input features are categorial. With this, WLS-GNN and GIN do not seem to differ much at all. * Although the experiments are conducted on a wide range of tasks, its comparison to recent methods feels quite shallow. For example, none of the presented baselines in Table 1 tackles the problem of graph classification. Why are authors not comparing against graph kernels/GNNs that are also evaluating on these graph classification benchmark datasets? Furthermore, it seems like the graph regression experiments on ZINC are not really comparable, since baselines do not seem to make use of edge features, while WLS does. [1] Xu et al.: How powerful are Graph Neural Networks?

Correctness: yes

Clarity: yes

Relation to Prior Work: I feel that this is not really the case. There are a lot of GNNs that perform transform-sum-cat like aggregations, and, in particular, the GIN formulation is mostly identical to the presented solution in case an MLP is added on top of the input node features.

Reproducibility: Yes

Additional Feedback: * From reading the paper, it is not clear to me how the presented approach can handle the problem illustrated in Figure 2(b). I feel like the results of Appendix B.3 should also be discussed in the main paper. * How is the readout function of the WLS-GNN defined? Is it the same as the one used in the WLS kernel? ============= Comments after Rebuttal ================== I thank the authors for their informative rebuttal. I still feel that this is an interesting paper regarding kernel theory, but my rating will remain unchanged. This is mostly due to the limited contributions regarding GNNs. For the revised manuscript, I highly recommend to give proper credit to the related work (Algorithm 1, COMBINE as concatenation, "transform-first-add-later" as described in GIN), and to also enhance the Graph Classification Table by the results reported in the related work.


Review 3

Summary and Contributions: Motivated by Weisfeiler–Leman test, this paper studied a novel way of basic embedding unit for graph neural network. This unit works in a transform-sum-cat fashion, which is in a different order from previous ones. Authors claimed that this method exploits the similarity preserving property in some metric space before and after embedding. The proposed method worked efficiently in experiments with less parameter size, and outperformed several selected counterparts in multiple tasks.

Strengths: The paper is well written and thus easy to follow. Theory in this seems rational and convincing. The proposed method is novel and I never see this in previous works. Graph neural network is of high relevance to several topic in machine learning, and thus is within the scope of Neurips.

Weaknesses: I mainly have the following concerns. 1) In general, this paper is incremental to GIN [1], which limits the contribution of this paper. While GIN is well motivated by WL test with solid theoretical background, this paper lacks deeper analysis and new motivation behind the algorithm design. I suggest the authors to give more insightful analysis and motivation. 2) I noticed that in Sec 5.3, a generator equipped with a standard R-GCN as discriminator tends to collapse after several (around 20), while the proposed module will not. The reason behind this fact can be essential to show the mechanism how the proposed method differs from previous one. However, this part is missing in this version of submission. I would like to see why the proposed module can prevent a generator from collapsing. 3) I understand that stochastic/random projection is with high probability to preserve the metric before mapping . My concern is that when stacking multiple layers of WLS units, the probability of the failure case of stochastic/random projection also increases (since projection is performed at each layer). This may greatly hinder the scenario of the proposed method from forming deeper GNN. In this case, authors should justify the stability of the proposed method. How stable is the proposed method? And what happens when stacking more layers?

Correctness: To me, claims in this paper seem to be appropriate and the experimental settings are correct.

Clarity: This paper is well written.

Relation to Prior Work: The contribution of the this paper compared to GIN is not well explained.

Reproducibility: Yes

Additional Feedback: I would consider to raise the rating if authors can address my concerns in the weakness part. ---after rebuttal--- I think the authors have addressed most of my concerns and other viewers'. While the reason of the collapse of the generator remains opaque, it might be difficult to give a thorough analysis during the rebuttal period. As such, I would like to raise my rating from 5 to 6.


Review 4

Summary and Contributions: This paper provides a set of tools for working on graphs that is different from existing work mostly by moving aggregation of adjacent node attributes from inside the loop to outside. Prior work operates on aggregated neighbor attributes and then learning is done on the aggregate, but this work learns on the neighbor attributes and then combines the attributes using concatentation. This allows finer comparison between graphs with different attributes but same adjacency. The contributions are - change in order of operations in computing node representation, a graph similarity based on this change and a demonstration of the use of this change as a graph neural network on some benchmarking data, synthetic and from biology/chemistry.

Strengths: The revisiting of order of operations is an interesting idea, and does provide benefits of distinguishing graphs of the roots of unity as described by the toy example in figure 2 that existing methods do not differentiate. I am not sure how many existing problems have that property where the sum of attributes sum to zero, so it might be artificial, but the empirical evaluation on the chemistry data set seems convincing. This approach does seem relevant to the segment of the NeurIPS community that works with graphs. I make no comment on the significance and novelty of the contribution and I am not that familiar with the prior art.

Weaknesses: There could be some cases where users of algorithms might have better generalization performance if the neighbor attributes were treated as a bag instead of separate entities like this method. The same way that linear SVMs sometimes generalized better than RBF kernels in the SVM era. I suppose in that case a linear kernel for the transform part would take the part of the aggregate, and then it should perform similarly to the aggregate first methods. You can sort of see the effect in Table 1 on the BZR and COX2 data set where WLSIn has a lower variance than WLS. Maybe a linear kernel baseline would help as well to convince the reader. Also since in line 235 the hyperparameters were tuned for each split, it's hard to tell if the method generalizes to an unseen split that wasn't tuned against.

Correctness: I'm a bit concerned about tuning each split individually as in line 235. It could lead to false security about generalization performance. In production system there is usually a secret test set that is not tuned against by the hyperparameter tuner to check for overfitting via the hyperparameter tuning process.

Clarity: Yes, although I am not familiar with the particular approach, the toy diagrams and clear explanations worked for me.

Relation to Prior Work: Yes, the idea of moving aggregation outside of the learning loop made sense to me.

Reproducibility: Yes

Additional Feedback: No change to my review post rebuttal, I find the paper interesting in terms of the kernel theory usage and stick to my score.

[Author Response · NeurIPS 2020]

We sincerely appreciate the reviewers for their careful reading, constructive questions and suggestions. We would very
much like further exchanges to improve our work, but the following is our best effort within the current limits.

First, we address questions appeared at least twice. We write **P1, P2** for paragraph reference, and **Rx** for reviewers.

**P1: Motivation of using similarity.** We discuss two main motivations here: lack of graph loss, and empirical failure
of distinguishing power. **First**, in tasks where the target is a fixed graph, e.g. graph autoencoding, we need an efficient
differentiable measure of graph similarity to guide the model toward the target. Existing works either use GNN without
principle or use costly graph matching. Being a graph pseudo-distance, WLS is a good candidate for graph loss; thus,
we successfully applied it to graph generation, although preliminary yet. **Second**, most of GNN theory, including GIN,
focuses on the distinguishing power. That is, they only consider when the representations are precisely equal. However,
higher distinguishing power (arXiv:1905.12560, 1905.11136) does not translate to empirical advantage. A reason we
suspect is that, deep learning works on continuous representations where distance can be more important than equality.
Optimization, generalization, and as **R1** suggested, adversarial robustness analysis of GNNs require the continuity
perspective. To the best of our knowledge, our work is the first to incorporate continuous similarity into designing GNN.

**P2: Comparison to GIN.** We believe WLS has sufficient advantages over GIN and other GNN models. **First,**
**difference in detail.** Both equation 4.1 in the GIN paper and official GIN implementation uses sum→weighted
sum→transform. The central node's ($v_c$) representation is distinguished from its neighbors' ($v_N$) only by scale. In
contrast, the WL test, and hence WLS, separate $v_c$ from $v_N$. Even with the simple WLS-GNN, the concatenation
makes the next MLP act differently to the pair. **Second, WLS makes different inductive biases easier to apply.**
WLS interprets the transformation as a feature map, and we experimented with two extreme biases: RBF kernel
without parameter learning, and MLP without significant inductive bias. Both cases show that WLS is capable of
generating useful graph representations, as shown from the empirical improvements. **Third, performance.** With no
other component than aggregation, WLS-GNN already outperforms popular models. Techniques from other GNN
models, e.g. pooling-over-layers from GIN or JKNet, may likely improve the predictive performance further. Moreover,
as **R1** kindly commented, we provided well-motivated reasoning for each step via the WL test and kernel distance.

The following are responses to the individual reviewers.

**Reviewer #1**. **Q1: Motivation of using similarity.** Please refer to **P1** for motivation. We appreciate very much the
suggestion of further subjects to study similarity-based models on. **Q2: Clarity.** We agree that a flow diagram can
significantly help the readers and will add one. We will move the GNN graph classification table to the main paper. **Q3:**
**Related work.** We will expand our review of GNN literature. Deep graph kernel is definitely in our scope and will be
mentioned. **Q4: Typos.** The comments are absolutely right and will be corrected. **Q5: Broader impact.** Thank you so
much for providing concrete examples. We will carefully discuss with our peers to consider further hazards.

Lastly, we are afraid of a slight difference in understanding that may affect the evaluation negatively. The mentioned
transform scheme for GNN is absolutely an interesting direction; however, we replaced the Taylor approximation and
random projection with an MLP, which results in a simple GNN similar to other models, yet empirically strong.

**Reviewer #2**. First, we apologize for including too much material in the appendix. **Q1: Compare to GIN and similar**
**aggregations.** Please see **P2** and additionally **P1**. **Q2: Graph classification with GNN.** The table is in the appendix,
but will be moved to the main paper. **Q3: ZINC.** Thank you for pointing this out. Only GatedGCN used edge features,
and we will rename it to GatedGCN-E. We also newly ran the WLS-GNN graph classification model without edge
features on ZINC. It obtained MAE 0.332±0.007, outperforming all baselines including GatedGCN-E significantly, but
not WLS-E. We will include it in the table. **Q4: Regarding Figure 2(b).** We appreciate the suggestion. We will move
the statements regarding Figure 2(b) to the main paper. **Q5: WLS-GNN readout.** We used the same readout used for
other models. A 3-layer MLP with specified dimensions is applied to the average of final layer node representations.

**Reviewer #3**. **Q1: Compare to GIN and explain motivation.** Please see **P1** and **P2**. **Q2: Why R-GCN collapses?**
WLS as a graph loss and a discriminator are our next subjects. Thorough investigation is required, but here is our
yet untested hypothesis. The MolGAN discriminator has several components, and it is unclear which graphs have
similar representations. In contrast, WLS creates similar representations for similar graphs, and possibly, chemically
valid graphs can be grouped together in the representation space, which helps stabilization. **Q3: Stacking random**
**projection may accumulate errors.** In downstream tasks, there are many other factors, but the accuracy degrades after
5-8 layers depending on the datasets. If we stack 10 projections with dimension 200, 95% of the norms are within 30%
error. For dimension 1000 we have 14% error. Considering many applications of shallow GCNs and the characteristics
of real-world graph datasets, we believe many tasks exist where this is sufficient. Details will be added to the appendix.

**Reviewer #4**. **Q1: Linear kernel.** Thank you for informing us of the issue. We tested linear SVM which was similar
to or marginally worse than the main paper on 5 out of 6 datasets, and significantly worse ($> 25\%$p difference) on
ENZYMES. The relative performance remains the same and will be mentioned. **Q2: About hyperparameters tuned**
**to each split.** We completely agree that one set of hyperparameters for all splits is the right way. Fortunately, despite the
variances across splits, the average accuracy is within $1\%$p between two tuning schemes, and we keep the conclusion.

[Meta-Review · NeurIPS 2020]

The reviewers liked the motivation and presentation of this new WL kernel-inspired GNN architecture, despite the actual architectural change to existing GNNs is relatively incremental. I hope the authors can revise their paper to address the reviewers’ comments in the camera ready version, and give proper credit to prior work.